# Resistant Starch-Encapsulated Probiotics Attenuate Colorectal Cancer Cachexia and 5-Fluorouracil-Induced Microbial Dysbiosis

**DOI:** 10.3390/biomedicines12071450

**Published:** 2024-06-28

**Authors:** Jui-Ling Wang, Yu-Siang Chen, Kuo-Chin Huang, Chin-Hsing Yeh, Miles Chih-Ming Chen, Lawrence Shih-Hsin Wu, Yi-Han Chiu

**Affiliations:** 1Animal Testing Division, National Laboratory Animal Center, National Applied Research Laboratories, Tainan 744, Taiwan; jlw1975@narlabs.org.tw; 2Department of Microbiology, Soochow University, Taipei 111, Taiwan; fiona90508@gmail.com; 3Department of Veterinary Medicine, School of Veterinary Medicine, National Taiwan University, Taipei 10617, Taiwan; 4Holistic Education Center, Mackay Medical College, New Taipei City 25245, Taiwan; kchsports@mmc.edu.tw; 5Fecula Biotech Co., Ltd., Tainan 744, Taiwan; sandy@fecula.com.tw (C.-H.Y.); miles@fecula.com.tw (M.C.-M.C.); 6Graduate Institute of Biomedical Sciences, China Medical University, Taichung 404, Taiwan

**Keywords:** chemotherapy-enhanced inflammation, colon carcinoma-associated cachectic symptoms, microbiome

## Abstract

5-Fluorouracil (5-FU) is commonly used as the primary chemotherapy for colorectal cancer (CRC). However, it can lead to unwanted chemoresistance. Resistant starch (RS), which functions similarly to fermentable dietary fiber, has the potential to reduce the risk of CRC. The effects of RS on improving CRC-associated cachectic symptoms and 5-FU chemotherapy-induced microbial dysbiosis remain unknown. Female BALB/cByJNarl mice were randomly divided into four groups: one tumor group (with CT26 colonic carcinoma but no treatment) and three CT26 colonic carcinoma-bearing groups that were administered 20 mg/kg 5-FU (T+5-FU group), a probiotic cocktail (4 × 10^8^ CFUs) plus chemotherapy (T+5-FU+Pro), or resistant-starch-encapsulated probiotics plus chemotherapy (T+5-FU+RS-Pro). T+5-FU and T+5-FU+RS-Pro administration significantly suppressed tumor growth and activated apoptotic cell death in CT26-bearing mice. 5-FU-induced increases in inflammatory cytokines and NF-κB signaling were mitigated by the Pro or RS-Pro supplementation. A gut microbial composition comparison indicated that the abundance of intestinal bacteria in the T and T+5-FU groups decreased significantly, while the groups receiving Pro or RS-Pro maintained a greater abundance and healthy gut microbiota composition, suggesting that RS can reduce the microbial dysbiosis that occurs during 5-FU chemotherapy. The use of RS-Pro before chemotherapy should be considered for the regulation of chemotherapy-associated cachectic symptoms, inflammation, and chemotherapy-induced microbial dysbiosis.

## 1. Introduction

Colorectal cancer (CRC) is the third most common malignancy and a major cause of cancer-related death [1]. It has affected the greatest number of Taiwanese people in the last 10 consecutive years [2]. Surgery, chemotherapy, and target therapy are mainstream CRC treatments. 5-Fluorouracil (5-FU)-based chemotherapy remains the primary choice for treating CRC. 5-FU is a water-soluble fluorinated pyrimidine analog that inhibits pyrimidine metabolism and DNA synthesis. To prevent cell proliferation, 5-FU primarily inhibits the thymidylate synthase, thus catalyzing the conversion of deoxyuridine monophosphate to deoxythymidine monophosphate and blocking the thymidine formation required for DNA synthesis [3]. Despite its undeniable merits, 5-FU chemotherapy per se is not appealing because it often induces serious side effects, such as myelosuppression [4], intestinal mucositis [5], and gastrointestinal toxicities [6]. In the meantime, significant breakthroughs in disease control and long-term survival have not been observed with single-agent 5-FU-based therapy when treating patients with advanced CRC. 5-FU-based therapy is frequently compromised by the development of chemoresistance, and the overall response rate in advanced CRC patients remains limited to 10–15% [7]. Thus, 5-FU is often administered in combination with agents such as leucovorin to treat CRC, and the combination of 5-FU with leucovorin has reduced the risk of cancer recurrence and improved survival [8].

The most severe side effects of 5-FU chemotherapy are intestinal mucositis and gastrointestinal toxicity [5,6]. Mucositis of the intestine is characterized by increased inflammation of the crypts and villus destruction, causing mucosal tissue to become susceptible to infection, ulceration, and general debility [9]. Severe diarrhea is the major 5-FU-induced gastrointestinal toxicity, which predominantly affects the upper and lower gastrointestinal tract and is observed in 10–30% of patients receiving 5-FU chemotherapy [10]. Gut microbiota, reactive oxygen species (ROS), and proinflammatory cytokines such as interleukin-1β (IL-1β), IL-6, and tumor necrosis factor-α (TNF-α) have been implicated in 5-FU-induced mucositis and gastrointestinal toxicity [11,12]. Previous studies have reported that 5-FU induced an imbalance of gut microbes and subsequent inflammation, resulting in exacerbated intestinal mucositis. A drastic shift from commensal bacteria (*Bifidobacterium* and *Lactobacillus* spp.) to pathogenic bacteria (*Escherichia*, *Clostridium*, and *Enterococcus* spp.) can follow single-agent 5-FU-based therapy [13].

Dietary patterns seem to play a key role in shaping the dynamic composition and diversity of gut microbiota. Dietary fibers such as resistant starch (RS) are well established as important substrates for gut microbial fermentation. RS can escape digestion and reach the colon, where it undergoes saccharolytic fermentation by *Bacteroides*, *Lactobacillus*, and *Bifidobacterium*, resulting in the production of short-chain fatty acids (SCFAs), mainly butyrate [14,15,16]. Butyrate contributes to normal colon function and plays important roles in reducing intestinal inflammation, improving gut barrier function, and lowering the risk for CRC [17,18]. Studies investigating the anti-CRC effects of RS have yielded inconsistent results. Some studies on the adverse effects of RS in promoting colorectal carcinogenesis have been conducted, while some animal and preclinical studies have shown that RS has a chemoprotective effect on preventing colorectal tumor development. In vivo experiments suggested that RS reduced the incidence of tumors by enhancing damaged cell apoptosis in the distal and proximal colon in 1,2-dimethylhydrazine dihydrochloride (DMH)-treated rats [19]. Using an azoxymethane (AOM)-treated rat model, dietary RS for 3 weeks post-AOM treatment significantly reduced ACF formation, while high doses of dietary RS prior to AOM treatment promoted the formation of ACF [20]. Moreover, in an Apc1638N mouse model of sporadic CRC, RS supplementation for up to 5 months significantly increased tumor formation within the small intestine. However, aspirin could prevent the small intestine tumor-enhancing effects of RS, suggesting that reducing inflammation may lower CRC risk [21].

In addition to dietary fibers, the role of probiotic bacteria in preventing CRC and modulating gut microbiota has been evaluated in both animals and humans. Supplementation for 10 weeks with an admixture of *Lactobacillus acidophilus* (NCFM^®^), *Lactobacillus paracasei* (Lpc 37TM), *Bifidobacterium lactis* (Bi-04TM), *Bifidobacterium lactis* (Bi-07TM), and *Bifidobacterium bifidum* (Bb-02TM) in a rat model bearing chemically induced CRC led to the occurrence of neoplastic lesions, while probiotic supplementation decreased the formation of aberrant crypts and ameliorated tumor malignancy, enhancing the antitumor effect of 5-FU chemotherapy in colic segments [22]. A fecal shotgun metagenomic-oriented study reported that the administration of *L. gallinarum* inhibited colorectal tumorigenesis in *Apc*^Min/+^ mice and in azoxymethane/dextran sulfate sodium-treated mice, suggesting that the tumor-suppressing effect was attributed to indole-3-lactic acid (ILA), a metabolite generated by *L. gallinarum* [23]. Furthermore, in a CT26-induced BALB/c murine model of CRC, 5-FU treatment reduced the overall diversity and the community composition of the gut microbiota, while the abundance of opportunistic pathogens decreased after probiotic administration [24].

As one of the best types of fiber for promoting butyrate levels, the effects of RS on improving the chemotherapy-induced inflammatory response and colon carcinoma-associated microbial dysbiosis remain unknown. A better understanding of the impacts of probiotics and RS on inflammation responses and microbial flora is required to determine how best to use them to treat CRC. In this study, our primary objective was to examine whether probiotics or RS-encapsulated probiotics exert protective effects against the 5-FU-induced inflammatory response and inhibitory effects on colon carcinoma-associated cachectic symptoms. Additionally, we focused on the modulatory effect of probiotics and RS-encapsulated probiotics on the 5-FU-induced dysregulation of the microbiome as a more specific objective.

## 2. Materials and Methods

### 2.1. Preparation of Probiotics (Pro) and Resistant-Starch-Encapsulated Probiotics (RS-Pro)

Five probiotics, *Bifidobacterium longum*, *Lactobacillus casei*, *Lactobacillus rhamnosus*, *Streptococcus thermophiles* (Glac Biotech Co., Ltd., Tainan, Taiwan), and *Clostridium butyricum* (New Bellus Enterprises Co., Ltd., Tainan, Taiwan), were mixed and used as a probiotic cocktail (Pro) at a ratio of 1:1:1:1:0.5. This probiotic cocktail was encapsulated with resistant starch (RS-Pro) extracted from green bananas through gelatinization, retrogradation, and lyophilization.

### 2.2. Cell Culture

The mouse colon carcinoma CT26 cells (American Type Culture Collection, Manassas, VA, USA) were tested for the absence of Mycoplasma spp. cultured in RPMI-1640 medium (Thermo Fisher Scientific, Waltham, MA, USA) supplemented with 10% fetal bovine serum (Thermo Fisher Scientific) and maintained in a humidified incubator (Thermo Fisher Scientific) at 37 °C containing 5% CO_2_.

### 2.3. In Vivo Studies

The animal study received approval from the Institutional Animal Care and Use Committee (IACUC) at the National Laboratory Animal Center (NLAC), NARLabs, Taiwan (NLAC(TN)-109-D003). Twenty-four female BALB/cByJNarl (B/c) mice (aged 7 weeks, body weight 20.8–24.1 g) obtained from the NLAC NARLabs were housed in a specific-pathogen-free animal room maintained at a controlled temperature of 21 °C to 23 °C, relative humidity of 45% to 65%, and 12 h light/dark cycles, with food and water available ad libitum. Fifty thousand CT26 cells suspended in 200 μL of DPBS were inoculated into the right flank of the mice subcutaneously. Four days later (defined as day 0), the mice were randomly divided into four groups (Figure 1): tumor (T), chemotherapy (T+5-FU), Pro plus chemotherapy (T+5-FU+Pro), or RS-Pro plus chemotherapy (T+5-FU+RS-Pro) groups. The 5-FU-treated mice were injected intraperitoneally with 5-FU (20 mg/kg, Sigma-Aldrich, St. Louis, MO, USA) twice per week for 3 weeks. The Pro and RS-Pro groups were administered Pro (4 × 10^8^ CFUs) and RS-Pro, respectively, once daily for 22 days. During the experiment, the tumor volume of the mice was measured with a digital caliper, and the tumor volume (TV) was calculated as length × width × 0.5. Body weight and clinical signs were monitored twice weekly. Submandibular blood was collected before and after treatment on days −7, 8, and 22 for complete blood cell (CBC) counting using a hematology analyzer (IDEXX, ProCyte Dx, Westbrook, MA, USA). At the end of the study (day 22), the tumor, gastrocnemius medialis muscle, spleen, and colon of the mice were harvested, weighed, and stored for further examination.

### 2.4. Blood Sample Analysis

Blood samples were obtained from the submandibular vein 7 days before and 8 and 22 days after the CT26 colonic carcinoma inoculation. White blood cell (WBC), neutrophil, monocyte, and lymphocyte counts were measured in blood samples (0.5 mL) using a blood cell analyzer (Symex K-1000, Sysmex American, Mundelein, IL, USA).

### 2.5. Western Blot Analysis

Colon and tumor tissues were lysed with 0.5 mL of CelLytic M lysis reagent (Sigma-Aldrich) containing 1% phosphatase inhibitor cocktail (Sigma-Aldrich) and protease inhibitor cocktail (Sigma-Aldrich). Then, each sample was centrifuged at 13,000× *g* for 30 min at 4 °C. A rapid Coomassie Kit (Bio-Rad Laboratories, Hercules, CA, USA) was used to determine the total protein concentration. We subjected the cell lysates to 10% SDS-PAGE and then transferred them onto a PVDF membrane followed by incubation with specific primary antibodies against IKK-β (Affinity Biosciences, Cincinnati, OH, USA), p-IKK-β (Affinity Biosciences), IκB-α (iReal, Taipei, Taiwan), NF-κB (iReal), and GAPDH (iReal). Subsequently, the PVDF membranes were incubated with secondary antibodies for 1 h at room temperature. The internal controls were GADPH. Immunoreactive proteins were detected using the Immobilon™ Western Chemiluminescent HRP Substrate kit (Millipore, MA, USA). Images were captured, and the intensities of the protein bands were analyzed using LabWorks^®^ software (V4.5, UVP Inc., Upland, CA, USA), with results expressed in arbitrary optical density units.

### 2.6. ELISA Assay

A multiplex ELISA kit (DuoSet, R & D Systems, Minneapolis, MN, USA) was used to evaluate the levels of the cytokines, IL-6, IL-10, and TNF-α in the colon and tumor homogenates of the animals. The assay was performed according to the manufacturer’s instructions for colon homogenate.

### 2.7. RNA Extraction and Real-Time PCR

RNA was extracted from total tumors or liver tissues using an Rneasy Mini Kit (Qiagen, Germantown, MD, USA). Subsequently, cDNA was synthesized using M-MLV reverse transcriptase (Promega, Madison, WI, USA) and oligo-dT15 primers (Promega). Real-time PCR was performed on a Bio-Rad iCycler iQ system. The quantitative real-time PCR analysis was conducted in a 25 μL reaction containing 12.5 μL of iQ SYBR Green Supermix (Bio-Rad), 5 μL of cDNA, RNase-free water, and 100 μM of each primer. The resulting values were normalized to the GAPDH mRNA level. The oligonucleotide primers for mouse *Bax* (5′-GGATGCGTCCACCAAGAAG-3′ and 5′-CAAAGTAGAAGAGGGCAACCAC-3′), mouse *Bcl-2* (5′-TGTGGTCCATCTGACCCTCC-3′ and 5′-ACATCTCCCTGTTGACGCTCT-3′), mouse *caspase-3* (5′-GGAGATGGCTTGCCAGAA GA-3′ and 5′-ATTCCGTTGCCACCTTCCT-3′), mouse *caspase-8* (5′-ATCTGCTGTATCCTATCCCACG-3′ and 5′-AGGCACTCCTTTCTGGAAGTTAC-3′), and mouse *GAPDH* (5′-ACAATGAATACGGCTACAG-3′ and 5′-GGTCCAGGGTTTCTTACT-3′) were used according to previously published sequences.

### 2.8. DNA Extraction and Sequencing

The extraction of bacterial DNA was performed using a stool DNA Isolation Kit (Qiagen, Hilden, Germany) following the manufacturer’s instructions. Extracts were then treated with DNase-free RNase to eliminate RNA contaminants. The DNA yield and quality were measured by PicoGreen and Nanodrop (Thermo Fisher Scientific, Pittsburgh, PA, USA). Input genomic DNA (10 ng) was amplified via polymerase chain reaction (PCR). The barcoded fusion primers *341F* (5′-CCTACGGGNGGCWGCAG-3′) and *805R* (5′-GACTACHVGGGTATCTAATCC-3′) were used to amplify the V3 and V4 regions (341F-805R). The final purified product was then quantified using quantitative PCR (qPCR) according to the qPCR Quantification Protocol Guide (KAPA Library Quantification kits for Illumina 16S Metagenomic Sequencing Library), and the quality was checked using the Qubit 2.0 Fluorometer (Thermo Scientific) and an Agilent Bioanalyzer 2100 system. Next, paired-end (2 × 300 bp) sequencing was performed on the MiSeq platform (Illumina, San Diego, CA, USA). The raw data were uploaded as FASTQ files after the demultiplexing of paired-end reads. The sequencing results in the form of FASTQ files were uploaded to the MetaGenome Rapid Annotation Subsystems Technology (MG-RAST) server for analysis. Illumina metagenomic datasets are available on the NCBI under the Bioproject accession number PRJNA1114302.

### 2.9. Bioinformatics Analysis

The amplicon libraries were sequenced using the Illumina MiSeq platform (Genomics BioSci & Tech Co., New Taipei City, Taiwan). Paired-end reads (2 × 300 bp) were trimmed using Trimmomatics (v0.39) [25] and demultiplexed using an in-house script. Sequences from both ends of the 341F-805R primers were trimmed using Cutadapt (v3.3) with the following criteria: a read length of ≥150 bp and an error rate of 0.1 as the default. DADA2 (v1.12) [26], which includes filtering out noisy sequences (denoising), merging paired-end reads, and removing chimeras to extract amplicon sequence variants, was subsequently used for preprocessing. The operational taxonomic unit (OTU) number was determined by clustering the sequences from each sample using a 97% sequence identity cutoff, utilizing the QIIME2 Naive Bayes Classifier (v2019.10). The alpha diversity was analyzed by means of a species richness estimator (Chao1 and Observed features) and a species evenness estimator (Shannon and Simpson) using the R package “vegan”. Taxonomic abundance was calculated using the QIIME2 Naive Bayes Classifier (v2019.10) from the reads of each processed sample.

### 2.10. Statistical Analysis

All experiments were performed at least in triplicate. Statistical differences were analyzed using GraphPad Prism, version 8.0 for Windows (GraphPad Software, La Jolla, CA, USA) by performing a one-way ANOVA test. A *p* value less than 0.05 was considered to indicate statistical significance.

## 3. Results

### 3.1. Effects of Probiotics (Pro) and Resistant Starch-Encapsulated Probiotics (RS-Pro) in Combination with Chemotherapy on Spontaneous Apoptosis and Tumor Growth

In BALB/cByJNarl mice inoculated with murine colonic carcinoma CT26 cells, the final primary tumor volumes were measured as 792.7 ± 1018.3, 497.9 ± 503.1, 1416.8 ± 382.4, and 731.0 ± 340.3 mm^3^ for the tumor group (T), the chemotherapy treatment (T+5-FU) group, the Pro plus chemotherapy (T+5-FU+Pro) group, and the RS-Pro plus chemotherapy (T+5-FU+RS-Pro) group, respectively. Mice that received 5-FU chemotherapy showed a 37.2% decrease in tumor size (*p* < 0.01, T+5-FU group vs. T group, Figure 2A); however, mice that received Pro plus chemotherapy showed a 78.7% increase in tumor size (*p* > 0.05, T+5-FU+Pro group vs. T group, Figure 2A). At the time of sacrifice, the mean total tumor weights were determined as 4.79 ± 3.03 (median 4.68) g, 2.95 ± 1.68 (median 2.51) g, 3.58 ± 4.04 (median 1.62) g, and 1.75 ± 1.89 (median 1.22) g for the T group, the T+5-FU group, the T+5-FU+Pro group, and the T+5-FU+RS-Pro group, respectively. As a result, all treatments inhibited tumor growth despite no statistical difference compared to the T group (*p* > 0.05, Figure 2B).

The expressions of *Bax*, *Bcl-2*, *caspase 8*, and *caspase 3* in tumor cells were measured using qPCR. The level of *Bax* in tumor cells increased with the 5-FU or Pro plus 5-FU treatment, while the *Bcl-2* expression in tumor cells was the highest in the T+5-FU+Pro group (Figure 3A,B). Thus, the *Bax*/*Bcl-2* ratio was greater in the T+5-FU and T+5-FU+RS-Pro groups than in the T and T+5-FU+Pro groups (Figure 3C). qPCR analysis indicated that the mRNA levels of *caspase 8* and *caspase 3* were elevated in tumors treated with 5-FU, Pro plus 5-FU, or RS-Pro plus 5-FU compared to those in the T group (*p* < 0.001, Figure 3D,E). Our results showed that probiotics alone do not exhibit antitumor activity, whereas the combination of RS-encapsulated probiotics with 5-FU significantly enhanced chemosensitivity and chemotherapy-induced apoptosis in CT26 colonic carcinoma cells in mice.

Moreover, ELISA and Western blotting analysis demonstrated the effect of treatment on the inflammation-related proteins in CT26-bearing mice. Treatment with 5-FU and probiotics reduced the protein levels of anti-inflammatory IL-10 (Figure 4A) and noticeably increased the protein levels of proinflammatory IL-6 and TNF-α in tumor tissues (Figure 4B,C). Starch-encapsulated probiotic supplementation, however, significantly induced IL-10 expression compared with that in the T+5-FU+Pro group (*p* < 0.05, Figure 4A). Moreover, the consumption of starch-encapsulated probiotics decreased IL-6 and TNF-α protein expression. After we observed that proinflammatory cytokines were activated by 5-FU, we further explored the effect of 5-FU on the NF-κB pathway in tumor tissues. Since IKK-β serves as an upstream kinase of IκB in the NF-κB signaling pathway, we investigated the effects of probiotics and starch-encapsulated probiotics on IKK-β activation. As shown in Figure 4D, Western blot analysis revealed no change in IKK-β protein levels. However, 5-FU chemotherapy increased the phosphorylation level of the IKK-β protein (p-IKK-β) compared to that in the T group. Notably, treatment with Pro and RS-Pro significantly reduced the p-IKK-β protein levels compared to those in the 5-FU group. There was no observable increase in the NF-κB protein in the tumor tissue compared to that in the T mice. However, the degradation of IκB-α was observed in T+5-FU+Pro and T+5-FU+RS-Pro mice, suggesting that probiotics or RS-encapsulated probiotics may lead to the degradation of IκB-α and the activation of NF-κB in tumor tissues.

### 3.2. Pro and RS-Pro Mitigate Cachectic Symptoms in CT26-Inoculated BALB/cByJNarl Mice

We investigated the protective effect of probiotics or RS-encapsulated probiotics by observing cancer cachectic symptoms, for which body weight, splenic weight, muscle weight, blood cell profile, and the colonic inflammatory response were monitored. The changes in body weights among the tested animals are summarized in Figure 5A. The body weights of T, the T+5-FU, T+5-FU+Pro, and T+5-FU+RS-Pro mice were increased by 6.8%, 4.2%, 5.8%, and 3.6%, respectively, in due course. In fact, the tumors grew considerably quickly, and the increases in body weight was proportional to the increase in tumor weight. However, the growth rate was significantly retarded when 5-FU chemotherapy, probiotics plus chemotherapy, or RS-Pro plus chemotherapy was supplied.

The changes in spleen mass were relatively minor: 0.81 ± 0.17, 0.60 ± 0.11, 0.76 ± 0.24, and 0.63 ± 0.14% of body weight for the T group, the T+5-FU group, the T+5-FU+Pro group, and the T+5-FU+RS-Pro group, respectively (Figure 5B). On the other hand, tumor inoculation seemed to have a negative effect on the gastrocnemius medialis (*GM*) muscle weight, whereas 5-FU chemotherapy, probiotics plus chemotherapy, or RS-Pro plus chemotherapy increased muscle weight. As a result, 5-FU chemotherapy, probiotics plus chemotherapy, or RS-Pro plus chemotherapy reduced the tumor inoculation-induced muscle weight loss (Figure 5C).

Figure 6 summarizes the hematological and immunological parameters. As presented in Figure 6A, tumor inoculation caused a significant decrease in the total white blood cell (WBC) count (including absolute neutrophils, monocytes, and lymphocytes) on day 8 and day 22 after CT26 inoculation (*p* < 0.05). There was a minor trend toward higher WBC levels on day 22 after CT26 inoculation in the T+5-FU group, the T+5-FU+Pro group, and the T+5-FU+RS-Pro group than in the T group, suggesting that 5-FU chemotherapy, probiotics plus chemotherapy, or the RS-Pro plus chemotherapy can maintain a steady WBC count. However, on day 22 after CT26 inoculation, the %neutrophils were decreased in the T mice, but the %neutrophils were markedly increased in T+5-FU+Pro and T+5-FU+RS-Pro mice (Figure 6B). In contrast, the %lymphocytes were decreased significantly on day 22 after CT26 inoculation in T+5-FU+Pro and T+5-FU+RS-Pro mice (Figure 6C). The %monocytes were largely unchanged among the four groups (Figure 6D).

The effects of various treatments on colonic inflammatory cytokines in CT26-bearing mice are shown in Figure 7. 5-FU chemotherapy caused a noticeable reduction in the colonic protein levels of anti-inflammatory IL-10 (Figure 7A). Probiotic supplementation, however, significantly induced IL-10 expression compared with that in the T+5-FU group (*p* < 0.05, Figure 7A). Moreover, the colonic protein levels of proinflammatory IL-6 and TNF-α were significantly lower in T+5-FU, T+5-FU+Pro, and T+5-FU+RS-Pro groups than those in the T group (*p* < 0.05, Figure 7B,C), suggesting that consumption of probiotics or RS-encapsulated probiotics may inhibit the effect of cachectic inflammation in colon tissue.

### 3.3. Pro and RS-Pro Mediate the Structural and Functional Compositions of Gut Microbiota

To characterize changes to the gut microbiota following various treatments, 16S rRNA sequencing of fecal samples from CT26-inoculated BALB/cByJNarl mice was performed. The taxonomic profiles at the phylum level revealed that the gut bacterial community in the T mice was dominated by Bacteroidetes (50.96 ± 12.76%), Firmicutes (46.19 ± 14.50%), and Proteobacteria (1.22 ± 0.01%) (Figure 8A). In the 5-FU-treated mice, we found a significant increase in Bacteroidetes (57.97 ± 12.12%) and a decrease in Firmicutes (38.99 ± 12.15%) in the gut microbiota. In contrast, compared with T mice, RS-encapsulated probiotic consumption increased the abundance of Firmicutes but decreased the abundance of Bacteroidetes (Figure 8A). Given that both Firmicutes and Bacteroidetes are influential regulators of the human gut microbiota [27], the ratio of Firmicutes to Bacteroidetes (F/B ratio) acts as an index of intestinal homeostasis. When the F/B ratio was studied in CT26-inoculated BALB/cByJNarl mice, there was no statistically significant difference between the T group and the other groups; however, the F/B ratio tended to increase with probiotics or an RS-encapsulated probiotic supplement (the Pro treatment was more prominent). In contrast, the average F/B ratio (0.720 ± 0.254) of the T+5-FU group was lower than that of the other groups, and the difference was statistically significant between the T+5-FU and T+5-FU+Pro groups (*p* < 0.05; Figure 8B). All alpha-diversity indices (Chao1, Observed feature, Simpson and Shannon indices) were significantly different between the T+5-FU group and the other three groups (Figure 8C–F).

A heatmap from the hierarchical clustering analysis based on the top nine different class taxa shows intersample changes among the four groups. The relatively highly abundant taxonomic classes included *Coriobacteriia*, *Saccharimonadia*, and *Bacteroidia*, likely as a result of the T and T+5-FU treatments, while they decreased after oral administration of probiotics plus chemotherapy or the RS-Pro plus chemotherapy (Figure 9A). In contrast, the relatively less abundant taxonomic classes include *Bacilli*, *Verrucomicrobiae*, and *Desulfovibrionia*, dissimilar to the results of the T and T+5-FU treatments, which were reversed after the oral administration of probiotics plus chemotherapy or the RS-Pro plus chemotherapy. A heatmap from the hierarchical clustering analysis based on the top 15 taxa of different orders shows intersample changes among the four groups. The relatively highly abundant taxonomic orders include *Coriobacteriales*, *Saccharimonadles*, and *Bacteroidales*, likely a result of the T and T+5-FU treatments, while they decreased after oral administration of probiotics plus chemotherapy or the RS-Pro plus chemotherapy (Figure 9B). The abundance of the orders *RF39* and *Monoglobales* also increased in the T+5-FU treatment group compared with those in the other groups.

At the genus level, we used a heatmap derived from hierarchical clustering analysis on the basis of the top 21 genera to summarize intersample changes among the four groups. Our data demonstrated a significant shift toward a dysbiotic gut microbiome in the T+5-FU group compared with other groups (Figure 9C). We then selected six differentially expressed genera and quantified their expression changes. Moreover, the abundances of *Enterorhabdus*, *RF39*, *Erysipelotrichaceae*, and *Candidatus Saccharimonas* significantly increased in the T and T+5-FU groups, while the trend was reversed when probiotics or RS-encapsulated probiotics were supplemented. The relative abundances of *Oscillospiraceae uncultured* and *Mucispirillum* in the gut of mice were significantly decreased when the 5-FU chemotherapy was applied, while the relative abundance was restored after probiotics or RS-encapsulated probiotics were supplementation (*p* < 0.05, Figure 9D).

## 4. Discussion

RSs are naturally present in various botanical sources and are classified into four distinct types based on their origin and characteristics: RS1 (physically inaccessible starch), RS2 (digestion-avoidant starch), RS3 (retrograde starch), and RS4 (chemically derived starch). Proper selection among RS types is critical for efficacy in clinical trials, and further evidence thus is required to evaluate the effectiveness of specific RS in treating CRC. The RS used in this research was type 2 RS extracted from green bananas through gelatinization, retrogradation, and lyophilization. Experimental studies suggest that RS2 is of particular importance because it plays a significant role in maintaining intestinal function [28], decreasing the concentration of secondary bile acids in the feces [29], and modulating inflammatory responses and the gut microbiota, and it is linked to protection against colitis and CRC. However, there is a scarcity of human trials evaluating the effectiveness of RS2 in treating CRC, and the existing evidence is inconsistent. While an epidemiological study suggested that higher RS intake was significantly associated with reduced cancer incidence and all-cause mortality [30], a clinical trial found that supplementing a habitual high-fiber diet with 32 g/day of RS2 for one week had no effect on putative risk factors for colon cancer, with the exception of increasing stool weight and colonic fermentative activity [31].

In this study, we confirmed that administration of probiotics or RS-encapsulated probiotics alleviated tumor growth, increased spontaneous tumor apoptosis, and decreased tumor inoculation and 5-FU chemotherapy-induced cachectic symptoms and the colonic inflammatory response. These results agree with numerous previous reports focusing on probiotics [22,23,24]. However, compared with the probiotic cocktail, RS-encapsulated probiotics have better inhibitory effects on tumor growth, cachectic symptoms, and the colonic inflammatory response, which suggests that viable probiotic bacteria are the main drivers of these health benefits; however, nonviable prebiotics are used to support and extend probiotics. Although a short-term probiotic supplementation has been shown to enhance host health by competitively excluding and inhibiting the growth of pathogens and triggering cytokine synthesis, a lack of fermentable prebiotics can lead to the loss of certain bacterial species, loss of gut microbiome diversity, and reduced production of beneficial fermentation end-products such as SCFAs [32]. Due to the long fermentation process from probiotic bacteria into postbiotic metabolites, partially digested in the upper segments of the alimentary tract, the immediate beneficial effect of prebiotics has increased the demand for prebiotic products. Prebiotics include fermentable dietary and functional fibers, such as inulin, fructooligosaccharides, galactooligosaccharides, lactosaccharose, and resistant starch, which are selectively utilized by host microorganisms and confer a health benefit by nourishing the commensal gut microbiota [33]. Thus, research is needed to enhance the study of prebiotics and allow us to exploit certain RS-derived prebiotics to prevent cachectic symptoms, inflammation, and CRC progression.

Recent research has highlighted the key role of microbiota in maintaining intestinal homeostasis, which is closely associated with the development of CRC tumorigenesis [34]. Dysbiosis in intestinal disease is considered to be an increase in the abundance of *Bacteroidetes* phylum bacteria, a decrease in the abundance of *Firmicutes* phylum bacteria, and a decrease in the alpha diversity index [35,36]. In this study, we observed that the *F*/*B* ratio and alpha diversity index were significantly decreased in the T+5-FU group, suggesting that 5-FU chemotherapy greatly inhibited the growth of *Firmicutes* and promoted the growth of *Bacteroidetes*. The gut microbiome is thought to increase the toxicity of chemotherapy and establish resistance to it, especially 5-FU. Previous studies have reported a trend toward an increase in the relative abundance of *Bacteroidetes* in 5-FU-treated, in vitro cultured human microbiota [37]. The main energy sources of Bacteroides are host-derived glycans, while *Bacteroidetes*, which produce the glycan-degrading enzymes, potentially increase the risk of colitis in mice and negatively affect CRC prognosis [38,39]. Notably, dietary nutrients can bolster or suppress the efficiency of 5-FU by modulating the metabolism of the gut microbiome [40]. Consistent with these findings, our results raised the possibility that a probiotic or RS-encapsulated probiotic supplement significantly reverses 5-FU-induced microbiota disturbance.

Additionally, our study revealed that the abundances of the orders *Coriobacteriales* and *Saccharimonadales* increased in the T and T+5-FU mice, leading to significantly greater abundances of the classes *Coriobacteriia* and *Saccharimonadia*, while they decreased after oral administration of probiotics plus chemotherapy or the RS-Pro plus chemotherapy. The class *Coriobacteriia* represents one of the deepest branching lineages within the *Actinomycetota* phylum, while several species within the *Coriobacteriia* class have been implicated in human diseases such as ulcerative colitis [41]. At present, there are few reports about *Saccharimonadales*. The abundance of *Saccharimonadales* was negatively correlated with the expression levels of cadherin-11, IL-17α, and Toll-like receptor (TLR)-2 in an adjuvant-induced arthritis rat model [42]. Guo et al. reported that the abundance of *Saccharimonadales* in an AOM/DSS+HFD colorectal adenoma murine model increased significantly, which may be related to the formation of adenomas [43]. In contrast, the relatively less abundant taxonomic classes included *Bacilli*, *Verrucomicrobiae*, and *Desulfovibrionia*, as a result of the T and T+5-FU treatments. *Bacilli* are ubiquitous in nature but found in relatively high concentrations in plant-origin food products [44]. Metabolically, *Bacillus* species very actively produce useful enzymes and numerous antibiotics [45]. Moreover, due to their ability to form endospores, strains of *Bacillus* strains are good potential probiotics [46]. As a major contributor to the maintenance of mucin integrity, *Verrucomicrobiae* represents only a small proportion of the healthy adult gut microbiome. However, in a human αSyn-over-expressing mouse model of Parkinson’s disease, Gorecki et al. reported that an endotoxin associated with intestinal inflammation was associated with a reduced abundance of *Verrucomicrobiae*, which resulted in impaired intestinal integrity and chronic intestinal inflammation [47].

At the genus level, 5-FU treatment may reduce the abundances of some bacteria (including *Oscillospiraceae uncultured* and *Mucispirillum*) but increase those of other genera (including the relative abundances of *Enterorhabdus, RF39, Erysipelotrichaceae*, and *Candidatus Saccharimonas*). As a mucus-dwelling pathobiont, *Mucispirillum* encodes a wide spectrum of enzymes that interact with intestinal mucin and create a positive environment for intestinal bacterial growth; thus, *Mucispirillum* spp. are thought to be associated with gut permeability and intestinal barrier integrity [48]. Reports have shown that *Enterorhabdus* in mammalian intestines can degrade mucus and thrive on the mucus layer; thus, it is associated with inflammatory bowel disease (IBD) risk [49]. *Erysipelotrichaceae*, which belongs to the *Firmicutes* phylum, is enriched in the lumen of colorectal cancer patients [50] and animals with 1,2-dimethylhydrazine-induced colon cancer [51].

*Candidatus Saccharimonas* (formerly known as TM7) belongs to the superphylum *Patescibacteria* and is found throughout the mammalian oral cavity and gastrointestinal tract [52]. The expansion of *Candidatus Saccharimonas* was demonstrated to be connected with inflammatory diseases such as gingivitis [53] and high-fat diet-induced obesity [54]. In a colitis-associated carcinogenesis model, the intestinal damage and expansion of *Candidatus Saccharimonas* were notably alleviated by the probiotic VSL#3 and fructooligosaccharide-inulin-based resistant starch, suggesting that *Candidatus Saccharimonas* could influence the inflammatory response during the initiation and progression of CRC [55]. Consistent with these findings, dysbiosis of the microbiota is closely associated with cancer risk and chemotherapy treatment. Our results found that even though 5-FU chemotherapy alleviated tumor growth and increased spontaneous tumor apoptosis, 5-FU also induced an imbalance in the gut microbial community associated with cachectic symptoms and colonic inflammatory response. Our results raise the possibility that 5-FU-induced dysbiosis can be reversed by the Pro or RS-Pro supplementation, which shifts the gut microbiota toward a more favorable composition.

## 5. Experimental Limitations

Like any scientific work, this study has limitations that should be acknowledged. The present study demonstrated that compared to the probiotic cocktail alone, RS-encapsulated probiotics effectively inhibited the growth of colon cancer CT26 cells by inducing apoptotic cell death. However, accurately estimating the daily intake dosage of RS remains challenging, preventing a thorough exploration of its dose-dependent effects in our current study. Additionally, further in vitro and preclinical studies are necessary to elucidate the detailed mechanisms underlying these observed effects.

## 6. Conclusions

This research revealed that probiotics and RS-encapsulated probiotics combat colon carcinoma-associated cachexia symptoms and colonic inflammatory responses, as well as mitigate 5-FU chemotherapy-induced microbial dysbiosis through distinct mechanisms. While the probiotic cocktail alone did not exhibit significant anticancer effects, the RS-encapsulated probiotics achieved a more potent inhibitory effect by enhancing both chemosensitivity and chemotherapy-induced tumor cell apoptosis. Both Pro and RS-Pro attenuated cachexia symptoms induced by colon carcinoma and 5-FU chemotherapy by modulating tumor-derived proinflammatory cytokines and regulating the NF-κB pathway within the tumor tissue. Additionally, both Pro and RS-Pro alleviated 5-FU-induced gut dysbiosis by modulating the structure and diversity of the gut microbiota. Gut microbiota imbalance might be a potential mechanism for the prevention of malignant transformation by Pro or RS-Pro, which is significant for the diagnosis, treatment, prognosis evaluation, and prevention of colorectal cancer. Such valuable information will provide the starting point for more targeted future investigations into probiotic and resistant starch supplements and gut microbiota.

## Figures and Tables

**Figure 1 biomedicines-12-01450-f001:**
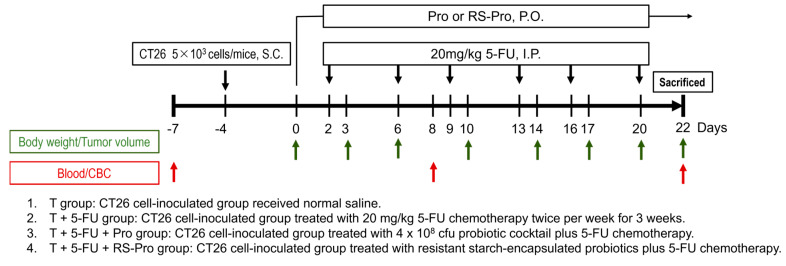
The treatment schedule for 5-FU chemotherapy, probiotics, and RS-encapsulated probiotics in CT26 tumor-bearing mice and analysis timelines.

**Figure 2 biomedicines-12-01450-f002:**
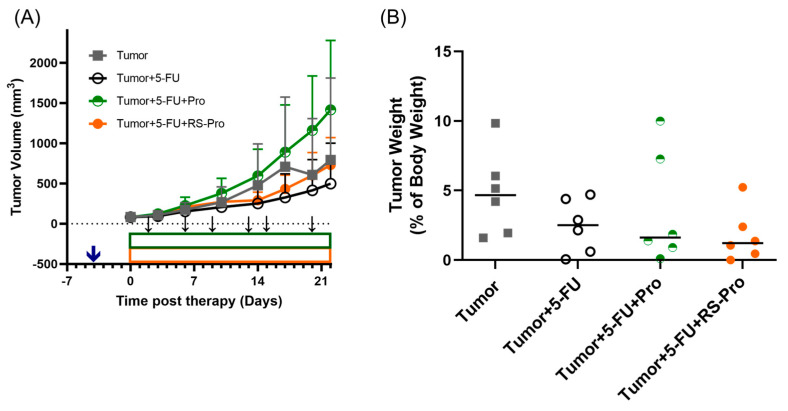
The impact of 5-FU chemotherapy, probiotics, and RS-encapsulated probiotics on tumor growth was assessed. (**A**) The tumor volume growth curve after various treatments. The blue arrow indicates CT26 cells were inoculated. (**B**) Tumor weights were measured at the study endpoint.

**Figure 3 biomedicines-12-01450-f003:**
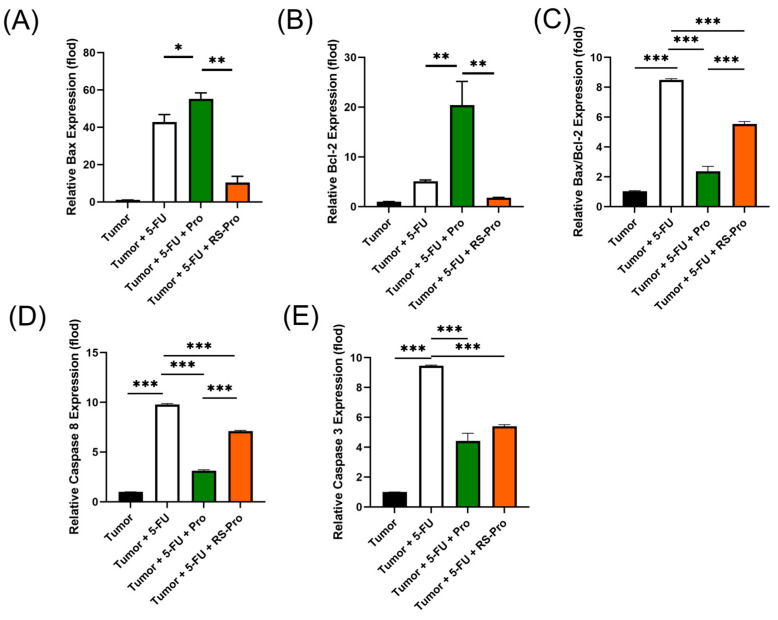
Quantification of mRNA levels of (**A**) *Bax*, (**B**) *Bcl-2*, (**C**) *Bax*/*Bcl-2* ratio, (**D**) *caspase 8*, and (**E**) *caspase 3* in the tumor tissues of mice treated with tumor inoculation, 5-FU chemotherapy, probiotics, and RS-encapsulated probiotics. The data are presented as means ± standard error of the mean (SEM). Statistical significance is denoted as follows: * *p* < 0.05, ** *p* < 0.01, and *** *p* < 0.001 when compared to the indicated group.

**Figure 4 biomedicines-12-01450-f004:**
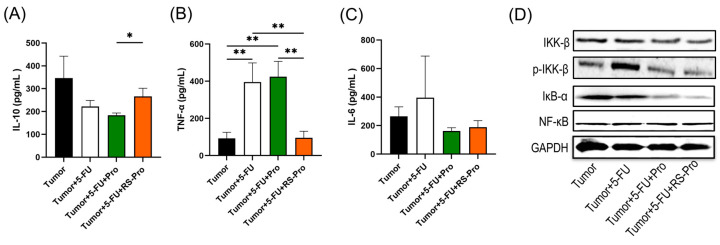
Effects of 5-FU chemotherapy, probiotics, and RS-encapsulated probiotics on the expression of inflammatory cytokines and NF-κB signaling in tumor tissue from CT26-inoculated BALB/cByJNarl mice. The protein levels of (**A**) IL-10, (**B**) TNF-α, and (**C**) IL-6 in the tumor tissue of mice were detected by means of ELISA. The data are presented as means ± SEM. Statistical significance is denoted as follows: * *p* < 0.05 and ** *p* < 0.01 when compared to the indicated group. (**D**) Semiquantitative analysis of colonic tissue protein levels of IKK-β, p-IKK-β, IκB-α, NF-κB, and GAPDH. GAPDH served as an internal control for equal loading.

**Figure 5 biomedicines-12-01450-f005:**
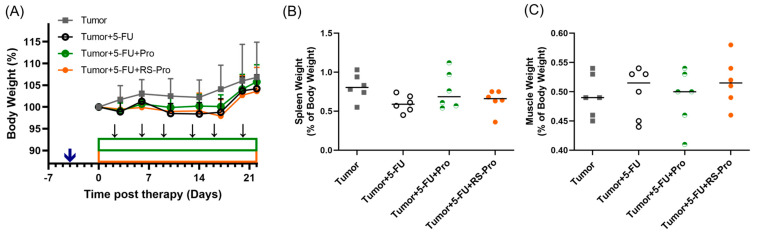
Cachectic symptoms after treatment with 5-FU chemotherapy, probiotics, and RS-encapsulated probiotics in CT26-inoculated BALB/cByJNarl mice. (**A**) The body weights of tumor-bearing mice were monitored throughout the treatment course. Data were presented as mean ± SEM. (**B**) Weight of spleen tissues. (**C**) Weight of GM muscle tissues.

**Figure 6 biomedicines-12-01450-f006:**
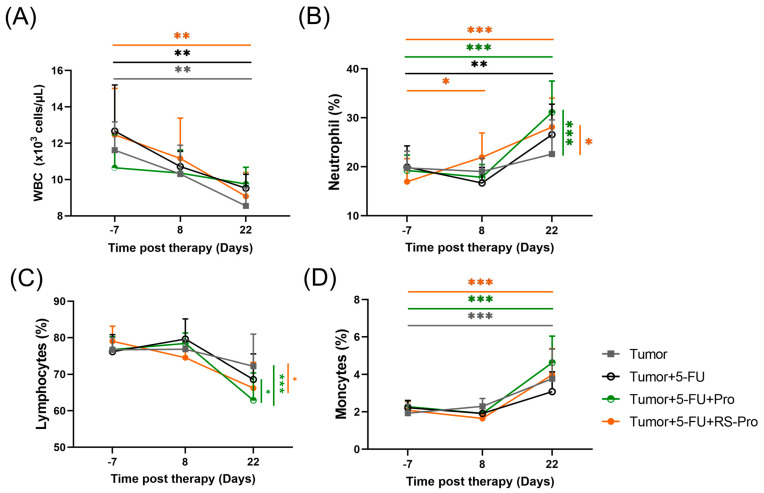
Changes in mean white blood cells (WBCs) (**A**), %neutrophils (**B**), %lymphocytes (**C**), and %monocytes (**D**) before (−7th) and the 8th and 22nd day after CT26 carcinoma were inoculated in BALB/cByJNarl mice. The data are presented as means ± SEM. Statistical significance is denoted as follows: * *p* < 0.05, ** *p* < 0.01, and *** *p* < 0.001 when compared to the indicated group.

**Figure 7 biomedicines-12-01450-f007:**
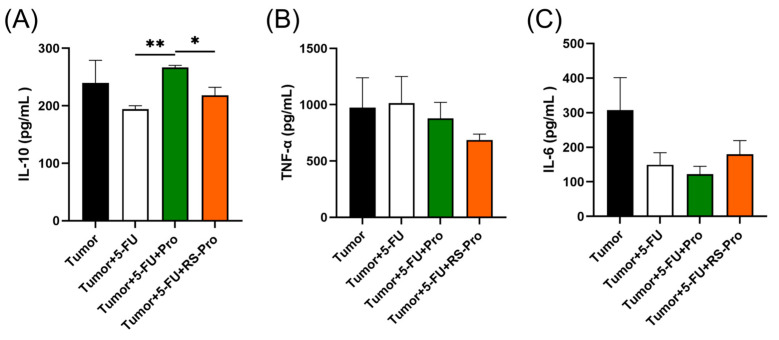
Effects of 5-FU chemotherapy, probiotics, and RS-encapsulated probiotics on the expression of inflammatory cytokines in colon tissue from CT26-inoculated BALB/cByJNarl mice. The protein levels of (**A**) IL-10, (**B**) TNF-α, and (**C**) IL-6 in the colon tissue of mice were detected by means of ELISA. The data are presented as means ± SEM. Statistical significance is denoted as follows: * *p* < 0.05, ** *p* < 0.01 when compared to the indicated group.

**Figure 8 biomedicines-12-01450-f008:**
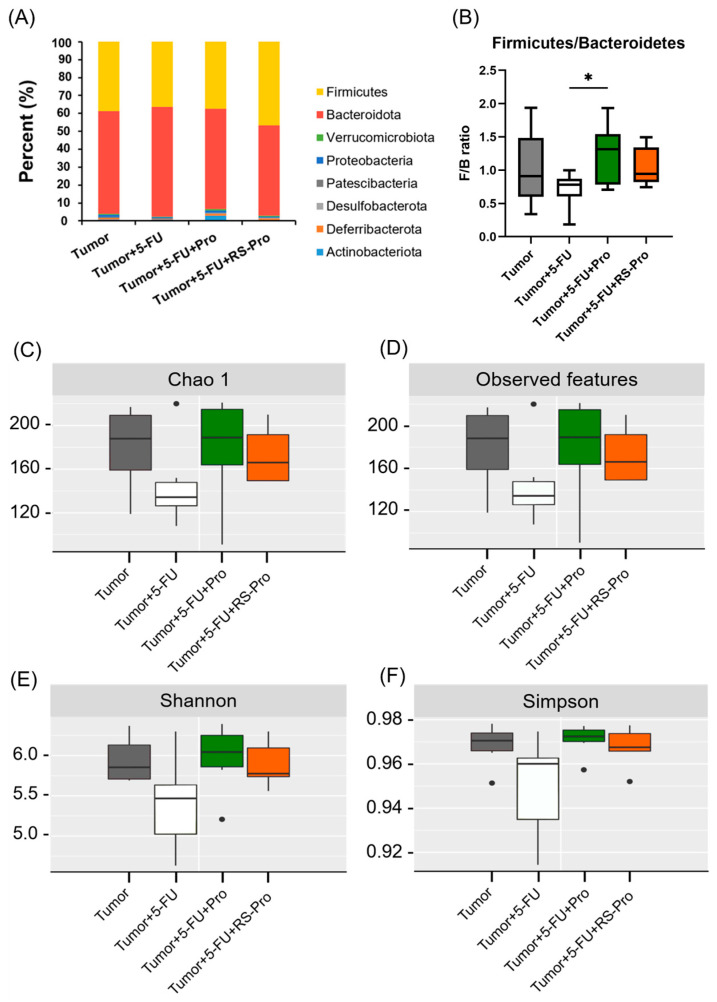
Effects of 5-FU chemotherapy, probiotics, and RS-encapsulated probiotics on the structural and functional composition of the gut microbiota. (**A**) Changes in the relative phylum-level abundances of gut microbiota components in CT26-inoculated BALB/cByJNarl mice. (**B**) The ratio of *Firmicutes* to *Bacteroidetes* (*F*/*B*) boxplot showing the gut microbiota in CT26-inoculated BALB/cByJNarl mice. Boxes contain 50% of all values, and whiskers represent the 25th and 75th percentiles. * *p* < 0.05 indicates a significant difference from the compared group. (**C**–**F**) Box plots showing the species richness estimator-Chao1 (**C**) and Observed features (**D**), and the species evenness estimator-Shannon (**E**) and Simpson (**F**) of alpha diversity.

**Figure 9 biomedicines-12-01450-f009:**
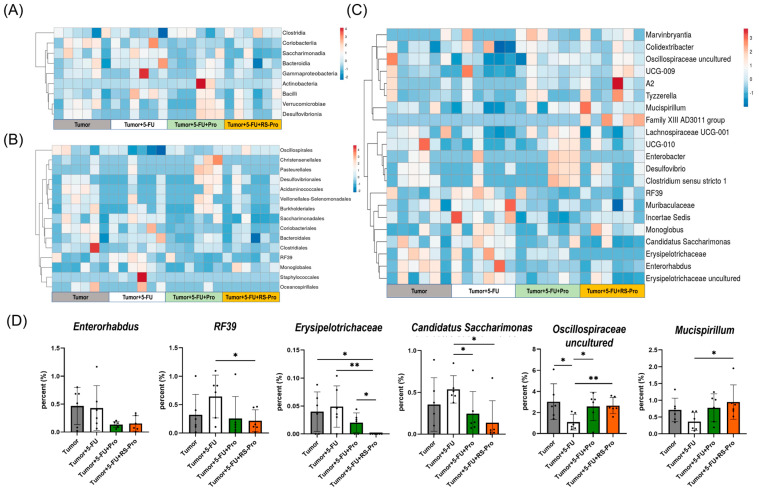
Heatmap depicting the relative abundance of the most abundant phyla (**A**), classes (**B**), and genera (**C**) (>0.1%) of the gut microbiota from different treatments. The color intensity of each sample is normalized to represent its relative ratio in CT26-inoculated BALB/cByJNarl mice. The colors from blue to red indicate the relative values of microbiota (−2 to 4). (**D**) The relative abundances of *Enterorhabdus*, *RF39*, *Erysipelotrichaceae*, *Candidatus Saccharimonas*, *Oscillospiraceae uncultured*, and *Mucispirillum*. The data are expressed as the means ± SEM. * *p* < 0.05, and ** *p* < 0.01 are significantly different from the compared group.

## Data Availability

The data used to support the findings of this study are included within the article.

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
