# Peer review of "Resistant Starch-Encapsulated Probiotics Attenuate Colorectal Cancer Cachexia and 5-Fluorouracil-Induced Microbial Dysbiosis"

_biomedicines, 2024, doi:10.3390/biomedicines12071450_

Round 1

Reviewer 1 Report

Comments and Suggestions for Authors

Abstract: more information about the dosage and specific results may improve the paper's readability.

Line 35-37: justify the sentence.

Line 61: ref. formatting is not in the journal requirement.

Line 112-115: looks like the conclusion of the study. At this stage, try to emphasize the objective of your study.

2.1: The concentration of each strain should be mentioned.

A scheme or figures detailing the experimental timeline and analysis will increase the easy understating of the study.

Why only NFkB and IkB-α is selected for evaluation? What about other players in NFkB pathways?

Is the selected study duration sufficient to attain a conclusion?

Correlated your results with the real-life situation. For example, what concentration of RS is good enough to support cancer patients, or how does the RS supplement prevent us from the onset of cancer?

Highlight the drawbacks or limitations of the study and future directions.

Similarly, it is about 36%; consider this while you revise the manuscript. 

Reviewer 2 Report

Comments and Suggestions for Authors

The manuscript titled "Resistant starch attenuates colon carcinoma-associated cachectic..." is truly a good piece of paper, which with minor revisions can be accepted. Below are my comments to the authors.

  1. Rephrase the title, as the current one sounds awkward.
  2. Modify the first two and the last sentence in the abstract.
  3. Are the keywords "resistant starch" and "5-fluorouracil" necessary?
  4. The last paragraph of the abstract definitely needs to be rewritten to clearly state the aim of the study.
  5. The sentence "Together, the antitumor activity of Pro and 255 RS-Pro plus 5-FU can be attributed to the apoptosis-promoting effect, where both..." needs to be rewritten for better clarity.
  6. Consider removing "However, the 269 degradation of I𝜅B-𝛼 was observed in T+5-FU+Pro and T+5-FU+RS-Pro mice, ..." as it might not be necessary.
  7. All figures are too small and almost unreadable. Please improve them.
  8. The Conclusion needs to be rewritten.

Overall, this is a very good paper.

Round 2

Reviewer 1 Report

Comments and Suggestions for Authors

The authors have improved the manuscript. Still, the similarity is more than 33%, which must be rectified before further processing. 
